# Tracing and analysis of 288 early SARS-CoV-2 infections outside China: A modeling study

**Francesco Pinotti**[1☯], **Laura Di Domenico**[1☯], **Ernesto Ortega**[2☯], **Marco Mancastroppa**[3,4], **Giulia Pullano**[1,5], **Eugenio Valdano**[1,6], **Pierre-Yves Boëlle**[1], **Chiara Poletto**[1], **Vittoria Colizza**[1]*

**1** INSERM, Sorbonne Université, Pierre Louis Institute of Epidemiology and Public Health, Paris, France, **2** Facultad de Física, Universidad de la Habana, Cuba, **3** Dipartimento di Scienze Matematiche, Fisiche e Informatiche, Università degli Studi di Parma, Parco Area delle Scienze, Parma, Italy, **4** INFN, Gruppo Collegato di Parma, Parco Area delle Scienze, Parma, Italy, **5** Sociology and Economics of Networks and Services lab at Orange Experience Design Lab (SENSE/XDLab) Chatillion, Paris, France, **6** Center for Biomedical Modeling, The Semel Institute for Neuroscience and Human Behavior, David Geffen School of Medicine, University of California Los Angeles, Los Angeles, United States of America

☯ These authors contributed equally to this work.
* vittoria.colizza@inserm.fr

**Data Availability Statement:** The database was made publicly available by the authors: "COVID-19 international cases as of Feb 13." https://docs.google.com/spreadsheets/d/1X_8KaA7l5B_

## Abstract

### Background

In the early months of 2020, a novel coronavirus disease (COVID-19) spread rapidly from China across multiple countries worldwide. As of March 17, 2020, COVID-19 was officially declared a pandemic by the World Health Organization. We collected data on COVID-19 cases outside China during the early phase of the pandemic and used them to predict trends in importations and quantify the proportion of undetected imported cases.

### Methods and findings

Two hundred and eighty-eight cases have been confirmed out of China from January 3 to February 13, 2020. We collected and synthesized all available information on these cases from official sources and media. We analyzed importations that were successfully isolated and those leading to onward transmission. We modeled their number over time, in relation to the origin of travel (Hubei province, other Chinese provinces, other countries) and interventions. We characterized the importation timeline to assess the rapidity of isolation and epidemiologically linked clusters to estimate the rate of detection. We found a rapid exponential growth of importations from Hubei, corresponding to a doubling time of 2.8 days, combined with a slower growth from the other areas. We predicted a rebound of importations from South East Asia in the successive weeks. Time from travel to detection has considerably decreased since first importation, from 14.5 ± 5.5 days on January 5, 2020, to 6 ± 3.5 days on February 1, 2020. However, we estimated 36% of detection of imported cases. This study is restricted to the early phase of the pandemic, when China was the only large epicenter and foreign countries had not discovered extensive local transmission yet. Missing information in case history was accounted for through modeling and imputation.

JPpwwV3js1L6lgCRa3FoH-gMrTy2k4Gw/edit?
usp=sharing.

**Funding:** This study is partially funded by the
Agence National de la Recherce (ANR, https://anr.
fr/) through the project DATAREDUX (ANR-19-
CE46-0008-03) to VC; the European Union with
grants RECOVER (H2020-101003589) and MOOD
(H2020-874850, https://ec.europa.eu/
programmes/horizon2020/en) to PYB, CP, and VC;
REACTing (https://reacting.inserm.fr/) through the
COVID-19 funding to VC; the Municipality of Paris
(https://www.paris.fr/) through the programme
Emergence(s) to FP and CP; INSERM-INRIA
partnership for research on public health and data
science to LDD. The funders had no role in study
design, data collection and analysis, decision to
publish, or preparation of the manuscript.

**Competing interests:** The authors have declared
that no competing interests exist.

**Abbreviations:** COVID-19, coronavirus disease
2019; RT-PCR, reverse transcription-polymerase
chain reaction; SARS-CoV-2, severe acute
respiratory syndrome coronavirus 2; STROBE,
Strengthening the Reporting of Observational
Studies in Epidemiology.

## Conclusions

Our findings indicate that travel bans and containment strategies adopted in China were
effective in reducing the exportation growth rate. However, the risk of importation was esti-
mated to increase again from other sources in South East Asia. Surveillance and manage-
ment of traveling cases represented a priority in the early phase of the epidemic. With the
majority of imported cases going undetected (6 out of 10), countries experienced several
undetected clusters of chains of local transmissions, fueling silent epidemics in the commu-
nity. These findings become again critical to prevent second waves, now that countries have
reduced their epidemic activity and progressively phase out lockdown.

## Author summary

### Why was this study done?

- Originating from China, COVID-19 outbreak has now become a global pandemic, with
  more than 4 million cases reported across all continents.

- Underdetection of imported cases from China in the early phase of the epidemic played
  a crucial role in the spreading of the virus across and within countries.

- We quantified importations over time in light of the implemented travel ban in China
  and assessed delay and rate of detection of the first imported cases responsible for seed-
  ing the epidemic across multiple countries.

### What did the researchers do and find?

- We collected information on all international cases outside China officially confirmed
  in the period from January 3 to February 13, 2020.

- We developed a statistical model to predict trends in importations and predicted a
  rebound effect in importations from South East Asia.

- By analyzing clusters of local transmission, we estimated the detection rate at 36%.

### What do these findings mean?

- Travel bans adopted in China contributed to reducing the growth rate of exportations;
  however, they did not prevent international seeding.

- The majority of imported cases went undetected, generating extensive chains of local
  transmission in countries outside China. This led to silently spreading epidemics in
  seeded countries.

## Introduction

After being first identified in China in January 2020, severe acute respiratory syndrome coro-
navirus 2 (SARS-CoV-2) reached all countries worldwide within a few months [1]. Massive

intervention measures [2] were implemented by Chinese authorities in late January to control the epidemic. Countries outside China promptly reinforced border controls and intensified active surveillance to rapidly detect and isolate importations, trace contacts, and isolate suspect cases [3,4]. Such a massive response did not prevent countries to face importations of cases and experience extended independent chains of local transmission [5–7], ultimately resulting in sustained local spread.

The effectiveness of measures aiming at preventing importations and local transmissions critically depends on disease epidemiology and natural history of the infection [8,9], as well as the volume of importations [3]. For the case of coronavirus disease 2019 (COVID-19), the presence of an incubation period, during which infected individuals carry on their usual activities (including travel), was a major challenge for screening controls at airports [8]. Moreover, mild non-specific symptoms and transmission before the onset of clinical symptoms [10–12] compromised infection control measures for importations and onward transmissions [9]. Imported cases went undetected and contributed to the global spread of the disease [13–18].

Here we systematically collected and analyzed data on the first 288 COVID-19 confirmed cases outside China, to characterize the international spread of COVID-19 pandemic during its early phase. We show that the detailed information carried by case histories provided early evidence of the COVID-19 specific features that enabled the virus to escape containment efforts and reach pandemic proportion. We analyzed importations that were successfully isolated and those leading to onward transmission, characterizing their case timeline. We developed a statistical model to describe trends in importations up to mid-February and quantified the proportion of undetected imported cases.

## Methods

### Data collection and synthesis

We collected all international cases confirmed by official public health sources in the period from January 3 to February 13, 2020. Case history was reconstructed by searching the scientific literature, official public health sources, and news [19–39]. Case history included dates of travel and symptom onset, date of COVID-19 confirmation, date of hospital admission, date of case isolation, travel history, epidemiological link with other cases, and hospitalization history. International cases included imported cases, secondary cases out of China, and repatriations. Cases from cruises were not considered here. Information was extracted by LDD and EO and checked by MM. Additional cases in the period from February 14 to February 27 were collected and used to validate the results.

The full database, along with the database describing clusters, was made publicly available [40].

This study is reported as per the Strengthening the Reporting of Observational Studies in Epidemiology (STROBE) guideline. STROBE checklist can be found in the Supporting Information (S1 STROBE Checklist). The study did not have a protocol.

### Descriptive analysis

For imported cases with full information on the timeline of events, we computed the average duration from travel to onset, from travel to hospitalization, and from hospitalization to reporting. We used analysis of variance to compare groups of imported cases that generated or did not generate local transmissions. We extended the analysis to all imported cases combining cases with full and partial information on the timeline. We used the analysis of variance and multiple imputation for missing data. Results were combined using Rubin's approach [41].

## Modeling and predicting importations

We modeled the total number of imported cases out of China over time accounting for date of travel, delay in reporting, and source areas.

We distinguished between 3 different sources: Hubei province (H), the rest of China (C), other countries (O). We modeled imported cases over time as a piecewise exponential function depending on the source and on travel restrictions in place. We assumed a different situation in Hubei province and the rest of the world because of the level of awareness in the different phases of the outbreak. The exponential functions are defined as follows:

$$I_{S,t} = \begin{cases} I_S^{pre} * e^{r_S^{pre}t}, & t \leq T_S \\ I_S^{post} * e^{r_S^{post}t}, & t > T_S \end{cases}, \quad S = H, \ C$$

$$I_{S,t} = I_S * e^{r_S t}, \ S = O$$

where $r_H^{pre}$ is the growth rate of cases coming from Hubei, and $r_C^{pre}$, $r_O$, with $r_C^{pre} = r_O$, the growth rates of cases coming from the rest of China and other countries, respectively. Travel restrictions were modeled by assuming a discontinuity in the growth rate. For Hubei, we assumed the growth rate to change from $r_H^{pre}$ to $r_H^{post}$ after the travel ban of January 23, 2020 (indicated with $T_H$); for the rest of China, we assumed an analogous change from $r_C^{pre}$ to $r_C^{post}$ after January 29, 2020 ($T_C$), date of first flight cancellations [42]. No change was considered for the other countries ($r_O$ constant over time), as no restrictions of travel were established toward these countries. The scale parameters of the exponential functions ($I_H^{pre}, I_H^{post}, I_C^{pre}, I_C^{post}, I_O$) were assumed to be different among the 3 sources, to account for different traveling volumes and dates of beginning of importations.

We modeled the time $\tau$ from importation to detection of a case with a gamma distribution, $g_t(\tau)$, conditioned to the date of case importation, $t$. The distribution $g_t(\tau)$ was assumed to have constant coefficient of variation (SD/mean) achieved by a constant shape parameter and a rate parameter varying smoothly in time to capture change in surveillance efficiency.

We used a Bayesian framework to fit the model to imported cases by origin, travel date, and confirmation date. Cases with partial information (e.g., missing date and/or origin of travel) were included by defining latent variables marginalized out during inference. The model was then used to predict imported cases 2 weeks in the future. All details of the analysis are reported in the Supporting Information (S1 Text).

## Estimation of underdetection of imported cases

We analyzed clusters of transmission generated by imported cases (index case(s) in each cluster) to estimate undetected importations. A cluster can be seeded by more than 1 index case, e.g., by infected family members who traveled together. The number of clusters of local transmission was modeled with a multinomial distribution according to the number of index cases and whether these were imported cases. As detailed in the Supporting Information, the likelihood function was a function of 4 observable quantities: the number of observed clusters with 1 imported index case ($x_1$); the number of observed clusters with more than 1 imported index case ($x_2$); the number of known imported cases causing no onward transmission ($\tilde{y}$); the number of clusters for which an index case was not identified ($z$); and a fifth and unobserved quantity, the number of undetected imported cases who did not start onward transmission ($w$). Maximization of the likelihood allowed to estimate $w$ from the records of $x_1$, $x_2$, $\tilde{y}$, $z$ and to compute the expected number of unobserved imported cases. Additional details are described in the Supporting Information (S1 Text).

## Results

### Timeline of travel-related cases

We collected 288 cases, including 163 imported cases, 109 cases involved in local transmissions, 30 repatriations, and 1 case of unknown origin. Fifteen cases were classified as both imported and local transmissions, since they contracted the infection outside China and traveled to a different country once infected (ES01, ES02, GB03, GB04, GB05, GB06, GB07, GB08, KR12, KR16, KR17, KR19, MY09, TH20, and TH21 in our database [40]).

Fig 1 summarizes the timeline of imported cases. Symptom onset occurred after the travel to the destination country for almost all cases for which date of travel and of onset are available (68 out of 73, 93%). Complete information was available for 51 (31%) imported cases, with quality of information decreasing over time (S2 Text, Fig A).

Among imported cases with full information, the delay from travel to hospitalization was longer in cases that generated secondary transmissions (mean of 10 ± 0.97 days compared

## timeline of imported cases

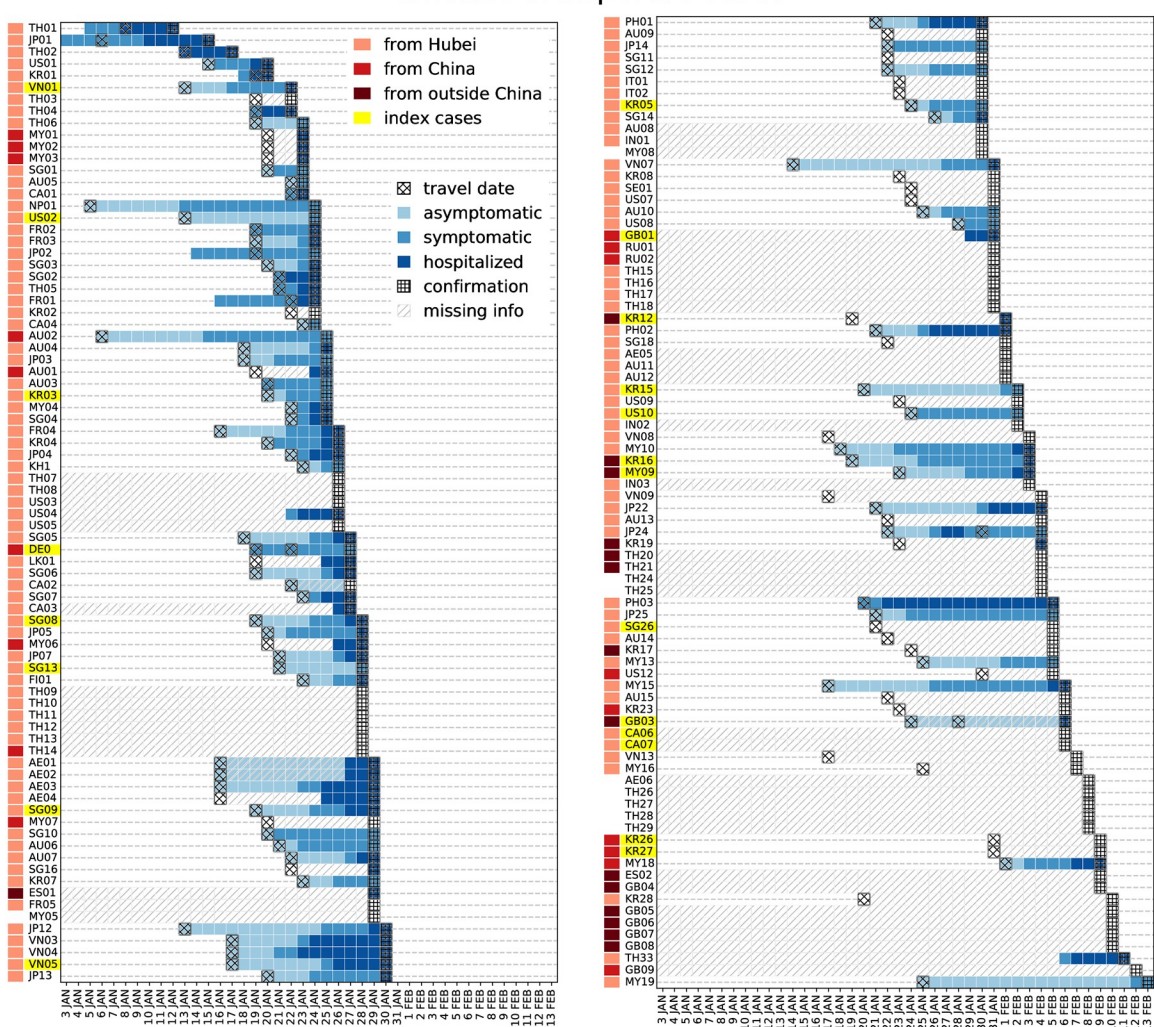

**Fig 1. Timeline of importation for all imported cases.** For each imported case, available information on travel date, onset date, hospitalization date, and confirmation date are displayed in the grid. Travel origin is color-coded (orange, red, brown for Hubei, China, outside China, respectively). Cases who generated a cluster upon arrival are highlighted in yellow.

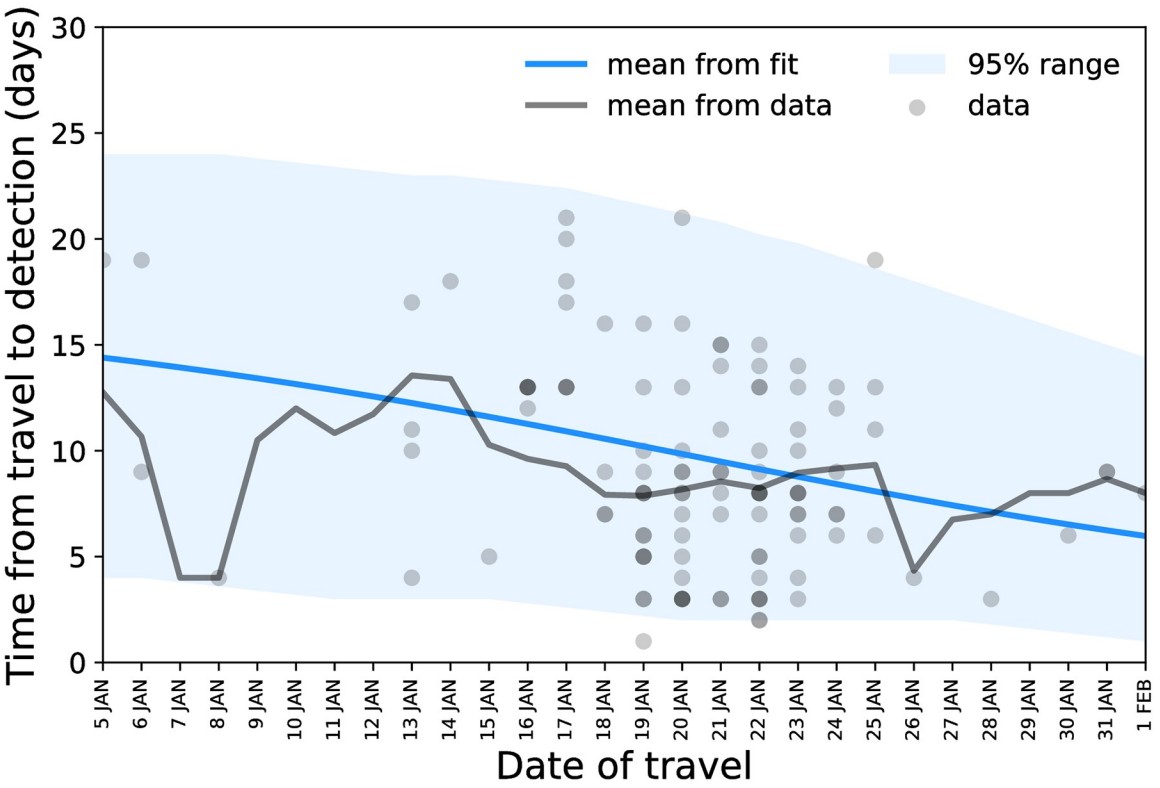

**Fig 2. Delay from travel to detection as a function of the date of travel: data points, mean, and model estimate.**

with 5.5 ± 0.67 days, $p$ = 0.003). Overall, the duration from travel to first event (whether symptom onset, or hospitalization for asymptomatic) was also longer, although the difference was not statistically significant (5.0 ± 0.9 days versus 3.7 ± 0.5 days, $p$ = 0.08). Durations of hospitalization were instead comparable among the 2 groups of cases (1.5 ± 0.7 days versus 2.6 ± 0.4 days for cases that generated or did not generate secondary transmissions, respectively). Including imported cases with missing information through imputation, we found the same trend though smaller in magnitude and not statistically significant (delay from travel to hospitalization 9.8 ± 1.2 versus 8.3 ± 0.5 days $p$ = 0.3; delay from travel to onset 5.8 ± 1.1 versus 4.2 ± 0.5 $p$ = 0.16, for cases that generated or did not generate secondary transmissions, respectively). This suggests that importations with missing information may be closer in characteristics to index cases leading to onward transmission.

The statistical model predicted a decrease in the average time from travel to detection from 14.5 ± 5.5 days on January 5, 2020, to 6 ± 3.5 days on February 1, 2020 (Fig 2).

## Predicting travel-related cases

The model predicted a rapid exponential growth of importations from Hubei, with a growth rate $r_H^{pre}$ = 0.26 (95% CI 0.21–0.31), corresponding to a doubling time of 2.8 days. In comparison, the exponential growth from other territories (rest of China and countries other than China) was slow, $r_C^{pre} = r_O$ = 0.04 (95% CI 0.00–0.08). After the implementation of travel restrictions, a negative growth rate was estimated, signaling a decline in imported cases. The decline was sharp for Hubei ($r_H^{post}$ = −0.64 [95% CI −0.85 to −0.48]) and more gradual for the rest of China ($r_C^{post}$ = −0.19 [95% CI −0.54, 0.00]). All estimated parameters and their confidence intervals are reported in the Supporting Information (S2 Text, Table B).

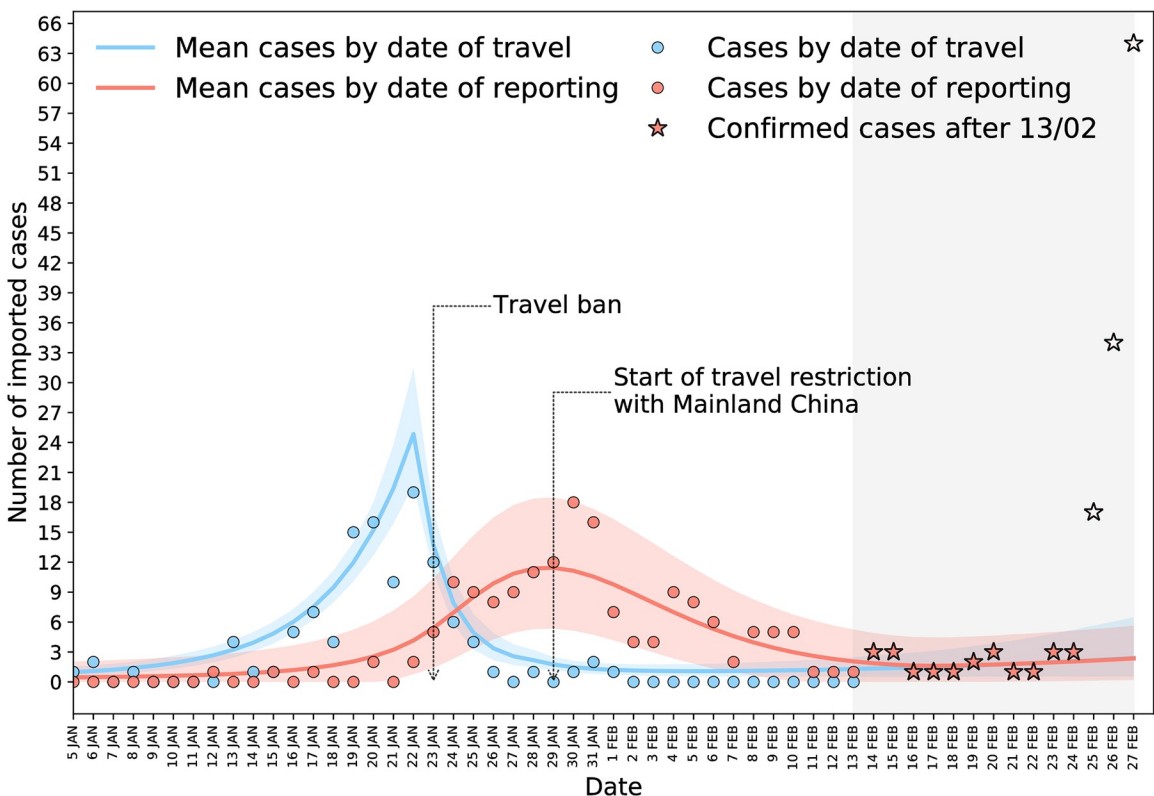

**Fig 3. Number of imported cases by date of travel and of reporting: data points and model predictions.** Stars correspond to model predictions in the successive 2 weeks; void stars refer to importations from Iran and Italy that could not be captured by the model because of its assumptions.

The predicted trend of all imported cases over time is shown in Fig 3, compared with the observed data. Reported importations were predicted to remain stationary in the second and third week of February and to rise again because of the effect of transmission clusters outside China. Imported cases after February 13, 2020, were in agreement with model predictions (Fig 3) up to February 24. Data in the successive 3 days diverged from our predictions as they were linked with early detection of multiple imported cases in the Middle East and Europe from sustained local transmission in Iran and Italy.

## Transmission clusters outside China

Forty-two transmission clusters were identified out of China in the timeframe under study. Table 1 summarizes the size and country of each cluster. Clusters were grouped according to whether the index case (1) was a traveling case identified prior to cluster detection, (2) a traveling case not identified or identified retrospectively once the cluster was observed, or (3) completely unknown. Assuming that clusters of unknown origin were linked to one of the already observed imported cases—or, in other words, not linked to an undetected imported case—led to an estimate of 76 (95% CI 49–118) undetected imported cases. In this scenario, detected cases would amount to 65% of all imported cases. Assuming instead that all clusters of unknown origin were due to undetected imported cases increased the number of undetected cases to 225 (95% CI 186–369), i.e., detected cases would correspond to only 36% of the total.

**Table 1. Summary of transmission clusters according to the type of index case.**

| Index Case | Number of Clusters | Clusters (size) |
|---|---|---|
| Traveler(s) identified prior to cluster detection[a] | 15 | cDE01 (16), cFR02 (12), cVN02 (7), cKR01 (5), cSG04[b] (5), cKR04 (3), cMY01 (3), cSG11[b] (3), cVN01 (3), cGB01 (2), cKR02 (2), cKR03 (2), cKR05 (2), cUS01 (2), cUS02 (2) |
| Traveler(s) not identified or retrospectively identified[c] | 8 | cSG01 (10), cSG02 (8), cJP01 (4), cCA01 (3), cKR06 (3), cTH04 (3), cFR01 (2), cJP02 (2) |
| Unknown[d] | 19 | cSG13 (8), cSG09 (5), cJP03 (3), cJP06 (3), cSG14 (3), cJP04 (2), cJP05 (2), cJP07 (2), cSG03 (2), cSG05 (2), cSG06 (2), cSG07 (2), cSG08 (2), cSG10 (2), cSG12 (2), cTH01 (2), cTH02 (2), cTH03 (2), cAE01[e] |

[a]The index case was identified independently from secondary transmissions.

[b]Cluster associated with 2 traveling cases.

[c]The index case was either identified retrospectively following case investigation prompted by the detection of secondary cases or the identity was not identified; however, the cluster was linked to a specific location/circumstance visited by Chinese travelers (shop, conference, bus tour).

[d]No connection with other case or source of infection has been identified yet.

[e]Insufficient information on the size of the cluster.

## Discussion

We reviewed here all confirmed cases out of China from January 3 to February 13, 2020, and gathered detailed information on case history and epidemiological links to (1) identify salient epidemiological features, (2) assess the impact of travel restrictions in China on the importations of cases worldwide, and (3) evaluate the effectiveness of control measures against importations. We found a rapid exponential growth of importations from Hubei, up to the closure of Wuhan airport preventing further travel of cases, combined with a slower growth from other countries in South East Asia. The estimated growth of importation before interventions is compatible with a doubling time of 2.8 days. Time from travel to detection considerably decreased across time, from $14.5 \pm 5.5$ days on January 5, 2020, to $6 \pm 3.5$ days on February 1, 2020. However, our estimates indicate that only 36% of imported cases were detected. This study is restricted to the early phase of the pandemic, when China was the only large epicenter and extensive silent transmission in other countries, such as Iran and Italy, was not discovered yet.

The substantially larger growth of importations from China compared with other territories is related to a stronger epidemic activity in the Hubei province, origin of the outbreak, with respect to other affected areas [5]. This difference is likely an outcome of the containment measures in China [2,43,44] and of the increased awareness following their implementation [45–49], leading to more efficient control. Identification, rapid management of cases, and contact tracing were indeed proposed by the World Health Organization as key to contain the epidemic globally.

We found that the travel ban in Wuhan produced a sharp decline in importations from the region. Combined with local containment measures in the rest of China and other countries in Southeast Asia, this resulted in a substantial overall reduction of exported cases worldwide. Indeed, after peaking at the end of January, registered traveling cases declined to plateau at very low levels. On one side, this shows that strict travel bans may be beneficial by reducing importations to manageable levels and by giving countries the time to prepare and strengthen their surveillance systems in the short term, as signaled by a reduction of the interval from travel date to detection over time. On the other hand, however, the decline likely occurred too late when local spreading was already established in many countries outside the epicenter of

the epidemic [50,51]. At the end of February, sustained community transmission in Iran and Italy led to exportations to several other countries in the world in a timeframe of few days [52,53]. This was facilitated by the fact that no local transmission was reported in these countries, preventing the alert to other countries for them to deploy targeted surveillance and control. At that time, indeed, case definition for the importation of a COVID-19 suspect case was based exclusively on China as the origin of exposure or travel [54,55], with few exceptions including East Asian countries [56,57].

Monitoring imported cases is critical in territories with sporadic epidemic activity where the attention is concentrated on preventing the introduction of cases. At the time of writing, countries such as China and Australia show limited local transmission and are concentrating strong efforts in border controls to avoid a second epidemic wave, that could be fueled by the large proportion of the population still susceptible to SARS-CoV-2. Strict measures, such as massive testing and quarantine of incoming travelers, are being implemented to block silent introductions of cases as it occurred at the beginning of the pandemic.

We found that during the first half of February, countries outside China witnessed an increasing reporting of clusters with no known epidemiological link [5,16]. Our estimates indicate a detection ability of 36% to ascertain imported cases in countries outside China. This means that approximately 6 imported cases out of 10 have gone undetected. Previous detection rates estimates range from 27% [15] to 38% [15,17], with variations across countries [13,15]. Ascertainment was estimated to be even lower (approximately 10%) when assessed on repatriations [58]. Here, we excluded from this analysis all repatriation events and cruises with outbreaks, as conditions for detection and identification may be different.

Underdetection may have been due to several different factors including asymptomatic infections, infections with mild clinical symptoms, health-seeking behavior and declaration of travel history, case definition, and underdiagnosis. A relative long prodrome phase preceding symptom onset (approximately 2 days [59]) may have limited the tracing and isolation of contacts. We found, indeed, that traveling cases generating transmissions following importation were, in general, spending more time in the community prior to hospitalization—although this result was diluted upon imputation of missing values—signaling that the period of mild or no symptoms during which the individual carries on a normal life is important for the generation of secondary cases. The epidemic emergency starting with one single critical case on February 21, 2020, in Italy and accumulating hundreds of cases in few days [16] showed that clusters had gone undetected and epidemiological links with the index case were not found. Subsequent developments of the pandemic showed that underdetection of importations was a clear pattern in fueling large-scale epidemics that spread unseen in several countries before the first critical cases were finally detected [52,60].

Our study is affected by limitations. Underdetection of imported cases during the early phase of the pandemic might have been higher than what estimated here, as our analysis is conditional to the identification of clusters of cases. Underdetection may also proceed from the imperfect characteristics of RT-PCR (reverse transcription-polymerase chain reaction) tests used to identify infected cases. Some cases tested for SARS-CoV-2 could have been falsely negative, and this would affect both analyses presented in the manuscript. This is in line with our conclusion that a large part of imported cases may have been undetected. As the number of detected imported cases grew considerably, the quality of the information on case history degraded over time. We used modeling and imputation to account for the missing information regarding the most recent imported cases. Extension of this study beyond the early phase of the pandemic is limited by the strong spatiotemporal heterogeneities currently characterizing the pandemic. Our assumptions are based on a context of localized epicenter and border controls targeted against importations from China and countries in South East Asia who

experienced local transmission during the early phase. As the pandemic regime shifted toward a multiple delocalized epicenter in late February, additional travel bans were implemented worldwide to prevent importations from Europe and North America. This led to heavy disruptions of the flight network and a change of focus toward within-country transmission.

Our findings provide critical epidemiological understanding on the rate of importation and detection of cases during the early phase of the pandemic. Though effective in reducing international spread, the travel ban in Wuhan did not prevent the seeding of the pandemic in other countries, later becoming new epicenters of the pandemic. The epidemiological features of COVID-19 facilitated the silent spread of the epidemic in seeded countries. The lessons learnt during the early phase of the pandemic become critical again to prevent second waves now that countries have reduced their epidemic activity and progressively phase out lockdown.

## Supporting information

**S1 Text. Statistical methods.** Modeling traveling cases, delay from arrival to detection and index case detection probability.
(PDF)

**S2 Text. Additional results.** Dataset of international cases, results of likelihood estimation, sensitivity analyses, and analysis of imported clusters.
(PDF)

**S1 Table. Official sources for international cases.**
(PDF)

**S1 STROBE Checklist.**
(PDF)

## Author Contributions

**Conceptualization:** Pierre-Yves Boëlle, Chiara Poletto, Vittoria Colizza.

**Data curation:** Laura Di Domenico, Ernesto Ortega, Marco Mancastroppa.

**Formal analysis:** Francesco Pinotti, Laura Di Domenico, Ernesto Ortega, Pierre-Yves Boëlle.

**Methodology:** Francesco Pinotti, Pierre-Yves Boëlle, Chiara Poletto.

**Supervision:** Chiara Poletto, Vittoria Colizza.

**Validation:** Giulia Pullano, Eugenio Valdano.

**Visualization:** Francesco Pinotti, Laura Di Domenico, Ernesto Ortega.

**Writing – original draft:** Pierre-Yves Boëlle, Chiara Poletto, Vittoria Colizza.

**Writing – review & editing:** Francesco Pinotti, Laura Di Domenico, Ernesto Ortega, Marco Mancastroppa, Giulia Pullano, Eugenio Valdano, Pierre-Yves Boëlle, Chiara Poletto, Vittoria Colizza.

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
