## [Editor Report · Decision Letter 0]

28 Feb 2020

Dear Dr Colizza, 

Thank you for submitting your manuscript entitled "Lessons learnt from 288 COVID-19 international cases: importations over time, effect of interventions, underdetection of imported cases" for consideration by PLOS Medicine.

Your manuscript has now been evaluated by the PLOS Medicine editorial staff [as well as by an academic editor with relevant expertise] and I am writing to let you know that we would like to send your submission out for external peer review.

Kind regards,

Adya Misra, PhD,

Senior Editor

PLOS Medicine

---

## [Decision Letter · Decision Letter 1]

14 Apr 2020

Dear Dr. Colizza,

Thank you very much for submitting your manuscript "Lessons learnt from 288 COVID-19 international cases: importations over time, effect of interventions, underdetection of imported cases" (PMEDICINE-D-20-00657R1) for consideration at PLOS Medicine. 

Your paper was evaluated by a senior editor and discussed among all the editors here. It was also discussed with an academic editor with relevant expertise, and sent to independent reviewers, including a statistical reviewer. You will note that both reviewers and editors request additional data are used to validate your model. The reviews are appended at the bottom of this email and any accompanying reviewer attachments can be seen via the link below:

[LINK]

In light of these reviews, I am afraid that we will not be able to accept the manuscript for publication in the journal in its current form, but we would like to consider a revised version that addresses the reviewers' and editors' comments. Obviously we cannot make any decision about publication until we have seen the revised manuscript and your response, and we plan to seek re-review by one or more of the reviewers.

We expect to receive your revised manuscript by May 05 2020 11:59PM. Please email us (plosmedicine@plos.org) if you have any questions or concerns.

We look forward to receiving your revised manuscript. 

Sincerely,

Adya Misra, PhD

Senior Editor 

PLOS Medicine

plosmedicine.org

Title-Please revise your title according to PLOS Medicine's style. Your title must be nondeclarative and not a question. It should begin with main concept if possible. "Effect of" should be used only if causality can be inferred, i.e., for an RCT. Please place the study design ("A randomized controlled trial," "A retrospective study," "A modelling study," etc.) in the subtitle (ie, after a colon).

Abstract- last sentence of the methods and findings section should be a limitation of your methodology

Introduction- please update the background information as new evidence has come to light since your submission

Methods- since more data are now available- please update the international cases (as possible) in order to validate your model. Please email me at amisra@plos.org to discuss this, as needed

Please specify the sources of your data in the methods section providing citations instead of placing this in an SI file

Please present and organize the Discussion as follows: a short, clear summary of the article's findings; what the study adds to existing research and where and why the results may differ from previous research; strengths and limitations of the study; implications and next steps for research, clinical practice, and/or public policy; one-paragraph conclusion.

Discussion-please revise the language in this section esp lines 184-190, line 206-209

You may wish to expand on local transmission, underdetection and undiagnosis of cases worldwide here

Please replace epidemic with pandemic

Please comment on second waves of infections as imported cases are rising in China

Please ensure that the study is reported according to the STROBE guideline, and include the completed STROBE checklist as Supporting Information. 1 Please add the following statement, or similar, to the Methods: "This study is reported as per the Strengthening the Reporting of Observational Studies in Epidemiology (STROBE) guideline (SChecklist)."

Comments from the reviewers:

Reviewer #1: This is a review for "Lessons learnt from 288 COVID-19 international cases: importations over time, effect of interventions, underdetection of imported cases" by Pinotti et al. This paper presents results from statistical inference on a model of case importation along with responsive intervention in regards to the early stages of the COVID-19 outbreak. The research is novel and of interest to a potentially large audience but I believe that the paper needs revisions before it is ready for publication. 

1) The language in this paper needs to be reframed as analysis of the early stages of an outbreak rather than a description of an ongoing and changing situation. Current sections read somewhat like a news article rather than journal article. Many of these problems can be fixed by changing tense. 

----

23-24 "With the majority of imported cases going undetected (6 out of 10), countries should be prepared for the possible emergence of several undetected clusters of chains of local transmissions."

184  "The reduced volume of exported cases worldwide following the travel ban may have given countries the time to prepare and strengthen their surveillance systems, as signaled by a reduction of the interval from travel date to detection over time."

192  "...from East Asian countries [29]. ECDC and WHO currently base their case definition on travel from China only [30,31], but this may rapidly change in the next days."

2) There are not enough references from primary sources - this paper should be framed around existing scientific knowledge from previous outbreaks. About a third of the references are from primary sources (published articles or books), another third are from organisational reports or medRxiv and the last third are from online sources (e.g. Twitter, online news). Additionally, a couple citations appear to be incomplete.

----

S. Bhatia et al., "Report 6: Relative sensitivity of international surveillance," p. 6.

3) Several sections were difficult to read or had grammatical errors.

----

13 "We characterized importations timeline..."

76 The definition in this section is confusing. Here S = H, C while in the supplementary information (which is clearer) S = H, C, O. Perhaps include S on the second line of the piecewise equation where S = O?

86 It is hard to read subsequent sentences where one ends and the other starts with a variable or function name. 

98-101 In "Estimation of under-detection of imported cases" it is not clear what you are saying here or how the multinomial is defined. The next part of the section (where you define the likelihood) clarifies but the intro in lines 98-101 is confusing:

"We modelled the number of such 'cluster seeds', i.e. groups of index cases, with a multinomial distribution depending on the portion of cluster seeds of size 1 or greater than 1 (for simplicity, this was taken as 2), on the probability of detection of a seed, and on occurrence of secondary transmission."

197-198 What has an ability of 36%?

"Our estimates indicate an ability of 36% to detect imported cases in countries outside China."

199 This reads poorly (perhaps "Previous detection rate estimates..." ?)

"Previous estimates range from 27% [13] to 38% [13, 15] detection rates, with variations across

200 countries [13, 15]."

4) Supplementary Information and Methods

----

There is a section with a sensitivity analysis where the ban date in Wuhan is a day later. It would be nice to have model comparison using LOO or WAIC between a few select models (piecewise vs single exponential, etc...) Your model should explain the data better than a naive model and it would be easy to show here. 

Fig S2: 

a) The MCMC chains appear to be thinned; this should be noted in the caption. 

b) You should state if the mixing in STAN reported any divergence if you are showing convergence of the chains.

c) You state that both the black line and histogram show the posterior distribution, but the black lines look a bit more like priors to me. 

5) Possible additional data for analysis

----

I realise the current COVID-19 pandemic is ongoing and any additional analysis will not be comprehensive. But given that there is onging data, the paper would benefit from comparing model predictions to a more current set of data. 

Reviewer #2: Pinotti and colleagues have developed a mathematical model, using a Bayesian inference framework, to forecast trends in imported cases in China and estimate the percentage of imported cases and changes rate of detection of these cases. They used available data on 288 COVID-19 cases from a variety of data sources. Two growth functions are used: one for China - split by hubei and rest of China and separate growth function for other countries, with a break in growth rate denoted by Ts when restrictions were put in place. A separate growth function is used for other countries but assumes no discontinuity in growth rate. The mathematics are fully documented in supplemental materials and will seem complex but fundamentally - this is largely modelling growth functions with a structured break due to travel bans over a period of time. Latent variables were used to incorporate missing data where cases had incomplete information, authors used an inference technique to marginalise ("summing out" the probability of a random variable given the joint probability distribution). The Bayesian framework means that there are quite a few distributions to fit around model parameters. A slight nit-pick of this would be that the supplemental could potentially better rationalise the authors assumption no the choice of distribution. It may seem familiar to those with mathematical modelling experience, but it would not likely to be obvious to most. For instance, gamma distribution skewed nature to reflect change in surveillance; also rationale needed to truncated the gamma distribution on the detection delay to 25 days. Apart from some minor text issues (Line 25 in Supplemental - I think the authors meant "tuples" here), I think the model is quite reflective of what is occurring currently. Empirically, the model fits the data very well. The rebound in imported cases is occurring in China (and other areas) is now occurring so it would be interesting to conduct some follow-up work given how fast paced new data is coming in. One final issue is potentially, which the authors do not address in the discussion is the impact of inaccuracies of testing. Some studies have characterised multiple shedding routes (https://www.ncbi.nlm.nih.gov/pubmed/32065057?dopt=Abstract). Accurate testing methods are work in progress but it's worth noting that when new studies do come out on these parameters, it would be worth incorporating potential false negative results which may have on the impact of rebounding infection rates from imported cases. Apart from these comments - I think this study is important and merits publications.

[LINK]

---

## [Editor Report · Decision Letter 2]

27 May 2020

Dear Dr. Colizza,

Thank you very much for re-submitting your manuscript "Lessons learnt from the first 288 COVID-19 international cases: a modelling study" (PMEDICINE-D-20-00657R2) for review by PLOS Medicine.

I have discussed the paper with my colleagues and the academic editor and it was also seen again by xxx reviewers. I am pleased to say that provided the remaining editorial and production issues are dealt with we are planning to accept the paper for publication in the journal.

[LINK]

We look forward to receiving the revised manuscript by Jun 03 2020 11:59PM. 

Sincerely,

Adya Misra, PhD

Senior Editor 

PLOS Medicine

plosmedicine.org

Requests from Editors:

Please adapt the title to better match journal style. We suggest: "Tracing and analysis of 288 early SARS-CoV-2 infections outside China: a modeling study".

Please add some additional quantitative details of interest to the "methods and findings" subsection of your abstract. For example, the observation noted at lines 177-178 might be worth quoting; likewise the doubling time mentioned at line 185.

At line 37, please begin the sentence "Our findings indicate that ..." or similar.

At line 183 and in other places, we are not sure that "Nowcasting" is a widely understood scientific term: please consider rephrasing. 

Early in the methods section of your main text, please state whether the study had a protocol or prespecified analysis plan (and if so attach the document as an attachment, referred to in the text). Please highlight analyses that were not prespecified. 

Please restructure the early part of the "Discussion" section of your main text: the first paragraph should summarize the study's main findings, with these being discussed in subsequent paragraphs.

Please remove the section on "Funding" from the end of the text (this information will appear in the metadata). 

In your reference list, please ensure that all citations meet journal style (italics should be rendered in plain text). For reference 9, for example, formatting should be as follows: Fraser C, Riley S, Anderson RM, Ferguson NM. Factors that make an infectious disease outbreak controllable. Proc Natl Acad Sci U S A. 2004;101:6146-6151.

Please rename the STROBE attachment "S1_STROBE_Checklist". In the "Section" column, please add paragraph numbers to accompany the section where individual items appear (avoiding line or page numbers, which generally change in the event of publication). 

Comments from Reviewers:

[LINK]

---

## [Editor Report · Decision Letter 3]

16 Jun 2020

Dear Dr. Colizza, 

On behalf of my colleagues and the academic editor, Dr. Richard Zachariah Aandahl, I am delighted to inform you that your manuscript entitled "Tracing and analysis of 288 early SARS-CoV-2 infections outside China: a modeling study" (PMEDICINE-D-20-00657R3) has been accepted for publication in PLOS Medicine. 

PRODUCTION PROCESS

PRESS

PROFILE INFORMATION

Thank you again for submitting the manuscript to PLOS Medicine. We look forward to publishing it. 

Best wishes, 

Adya Misra, PhD

Senior Editor 

PLOS Medicine

plosmedicine.org